# The Bm86 Discovery: A Revolution in the Development of Anti-Tick Vaccines

**DOI:** 10.3390/pathogens12020231

**Published:** 2023-02-02

**Authors:** Alina Rodríguez-Mallon

**Affiliations:** Animal Biotechnology Department, Center for Genetic Engineering and Biotechnology, Avenue 31 between 158 and 190, P.O. Box 6162, Havana 10600, Cuba; alina.rodriguez@cigb.edu.cu

**Keywords:** Bm86, anti-tick vaccines, ectoparasites

## Abstract

The presence in nature of species with genetic resistance to ticks, or with acquired resistance after repeated tick infestations, has encouraged the scientific community to consider vaccination as an alternative to the unsustainable chemical control of ticks. After numerous attempts to artificially immunize hosts with tick extracts, the purification and characterization of the Bm86 antigen by Willadsen et al. in 1989 constituted a revolutionary step forward in the development of vaccines against ticks. Previously, innovative studies that had used tick gut extracts for the immunization of cattle against *Rhipicepahalus microplus* (previously named *Boophilus microplus*) ticks, with amazingly successful results, demonstrated the feasibility of using antigens other than salivary-gland-derived molecules to induce a strong anti-tick immunity. However, the practical application of an anti-tick vaccine required the isolation, identification, and purification of the responsible antigen, which was finally defined as the Bm86 protein. More than thirty years later, the only commercially available anti-tick vaccines are still based on this antigen, and all our current knowledge about the field application of immunological control based on vaccination against ticks has been obtained through the use of these vaccines.

## 1. Introduction

Today, the main method used to control ectoparasite infestations in livestock and companion animals is still the application of chemicals. A whole range of these substances, such as organochlorines, organophosphates, amidines, and pyrethroids, among others, have enabled tick control and eradication programs around the world [1]. More than 98% mortality can be achieved when the chemical used has a demonstrated efficacy against species living in a specific region. However, this efficacy means that there is a very small percentage of the population that will be randomly resistant to a given chemical, and, after a relatively short period of repeated intensive use of the same substance, and/or after a misapplication, a tick population resistant to that substance may have been selected, making it necessary to change to another kind of chemical [2,3]. In fact, resistant and multiresistant tick strains have been reported in many countries [4,5,6,7,8]. As concerns about food and environmental contamination with these chemicals are added to the above situation, the need to apply other, more sustainable, strategies to ectoparasite control is evident.

As early as the first half of the 20th century, natural or acquired resistance to tick feeding had already been well documented in both laboratory and domestic animals [9,10,11,12]. Those papers and some others showed that this resistance was mediated by a host immune response [13,14,15,16]. Artificial immunizations of hosts with extracts of salivary glands resulted in the induction of a host-protective response similar to that of acquired tick immunity characterized by hypersensitivity reactions [17,18,19,20]. However, it was also demonstrated that tick feeding could stimulate host antibodies against antigens other than those associated with salivary glands, because rabbits infested with *Hyalomma anatolicum excavatum* and *Rhipicephalus sanguineus* ticks also developed antibodies against digestive tract antigens [21].

A novel concept of “concealed” antigens able to stimulate an effective anti-tick response was introduced by Allen and Humphreys in 1979 [22]. In this pioneering work, antigens that are not presented to the host immune system during tick feeding, but are exposed to their effector elements through the blood meal were used to induce an effective immunity against ticks. This concept had previously been noticed following an increase in the death rate of *Anopheles stephensi* mosquitoes fed on rabbits immunized with a preparation of ground mosquito midgut [23]. Interestingly, this induced immunity against the internal organs of ectoparasites is quite different from that acquired naturally. The latter has an effect on the engorged female yield, whereas the vaccination-induced immunity results in reductions of the female engorgement weight and egg laying. The direct consequence of these effects is a reduction in the tick population [24].

The work published by Willadsen et al. in 1989 [25] brilliantly provided the long-awaited isolation and characterization of an antigen involved in the previous host immunizations with tick gut extracts that successfully limited *R. microplus* tick feeding and reproductive performance [26,27,28]. This important discovery allowed a deeper characterization of the mechanisms by which the host immunological response to concealed antigens exerts its effects on ticks [29,30,31], and also allowed the technological development of Bm86-based vaccines against *R. microplus* ticks [31,32,33,34,35,36,37]. Finally, this finding paved the way for research into new concealed antigens for anti-tick vaccine development [38,39,40].

## 2. Discovery

The paper published by Willadsen et al. in 1989 [25] described in detail the process of Bm86 purification from a crude membrane preparation of the Yeerongpilly strain of *R. microplus* ticks, using previous experience in the isolation of an antigenic fraction capable of conferring bovine protection against ticks [24]. Briefly, they used lectin affinity chromatography (wheat germ lectin and Con A) after the preparation of the membrane extracts, followed by preparative isoelectric focusing (IEF). The 5.1–5.6 PI range was pooled and subjected to serial HPLC gel filtration. The glycoprotein purified in this way, with a molecular weight of around 89,000 Da, was used as an antigen in two immunization experiments in bovines, which differed essentially in the quantity of antigen and the final formulation with or without adjuvant (CFA, IFA, or none). After three shots, given 4 weeks apart, bovines were challenged with 1000 tick larvae per day for 3 weeks (Figure 1). A 92% reduction in the number of larval progeny from ticks fed on vaccinated animals compared to controls was calculated, taking into account the effects of vaccination on the number and average weight of ticks recovered each day on individual cattle and a visual estimate of damaged ticks, and also the weight of eggs laid per gram of engorged ticks. These results confirmed that Bm86 was the molecule responsible for the damaged gut cells in ticks fed on hosts previously immunized with tick membrane extracts [24,41].

In addition to these essential experiments, the authors also described structural features of several peptides from the 182 amino acids sequenced of the Bm86 protein and demonstrated that this antigen was located on the tick gut digestive cell surface via indirect immunofluorescence, using either bovine antisera to the native Bm86 or rabbit antisera to a recombinant Bm86 protein produced in *Escherichia coli*. Finally, pre-incubations of tick digestive cell suspensions with antisera from vaccinated bovines showed a very strong inhibition of the endocytosis ability of fluorescein-labeled BSA in these cells. Heat-inactivated antisera were as inhibitory as unheated sera, which demonstrated that complement was not responsible for this inhibition. All these results together constituted indirect evidence suggesting putative biological functions for this protein which have still not been well elucidated more than thirty years later [42,43,44,45,46].

## 3. Impacts

Multicellular parasites are the most complex pathogens to be combated by vaccination, especially ectoparasites which are in contact with the host immune system only during feeding and, in addition, have developed highly effective methods for eluding the host immune system over millions of years [47]. Hence, the great scientific challenge of achieving ectoparasite control through immunization. However, the work developed by Willadsen et al. in 1989 [25] demonstrated that host vaccination with a defined hidden tick antigen can be used to induce an immunologic response whose effector mechanisms take place inside ticks after a blood meal. Unlike immunization with salivary gland proteins, which increases host cutaneous sensitivity to ticks, these internal molecules are not introduced into the host during feeding and the immune reaction does not occur at the host–parasite interface, which is a desirable characteristic of an anti-tick vaccine in order that unwanted side effects such as skin irritation may be avoided in its practical application [48,49]. However, the advantage implicit in the concept of concealed antigens requires multiple host immunizations in order to guarantee high antibody titers against the target molecule and consequent efficacy against ticks [33,50].

The isolation of Bm86 and the description of its amino acid sequence [25] allowed the cloning of its coding sequence and its recombinant expression in different hosts [32,51,52]. These biotechnological approaches have permitted the protein to be obtained in sufficient quantities to develop the only registered, commercial anti-tick vaccines, which have demonstrated efficacy against *R. microplus* ticks under field conditions [35,37,53,54,55]. These Bm86-based vaccines have also shown successful immunological efficacy against other tick species, expanding their practical use for tick control in other species [56,57,58,59,60,61,62,63]. However, there have been variable results with different strains and species of ticks [60,64,65,66]; even today, these differences lack a clear explanation [64,67]. Although this antigen has been used to immunize cattle for more than thirty years, its biological function in the gut membrane cells and the reasons for varying levels of protection against different ticks have not yet been completely elucidated.

Despite these gaps in knowledge, the implementation of the Bm86-based vaccine in large-scale production, with more than 3 million vaccinated cattle, has demonstrated the efficacy of this vaccine to control ticks [36,68,69], including pesticide-resistant tick strains [6,35]. These impressive results have made the Bm86 protein into the reference antigen for all studies on anti-tick vaccine development and stimulated the scientific community to search for new antigens with broader action spectra, to be used either alone or combined with Bm86 in tick control programs [38,40,70,71,72,73]. These investigations had a “boom” in the first fifteen years of the 21st century, as evidenced by the number of papers and presentations at international conferences addressing the topic [74]. Notwithstanding promising results obtained in laboratory experiments with diverse antigens against different tick species, none has been tested under field conditions, nor have any been registered as commercial products [75,76]. This fact reflects that the concept demonstration of an effective antigen against ticks is only the first step of a long and costly road to sanitary registration of an anti-tick vaccine, in which there are many involved actors.

On the other hand, the experience in the field application of Bm86-based vaccines has demonstrated that anti-tick vaccines are hard to commercialize because a change in thinking based on classic concepts of preventive vaccines against viruses and bacteria is needed. These are not “knock down” vaccines. The reduction in the tick’s reproductive capacity achieved by the Bm86-based vaccines leads to continuing reductions in tick populations after two or three generations of feeding on vaccinated animals, keeping their infestations at an acceptable level for livestock and favoring the enzootic equilibrium for hemoparasitic diseases. As a consequence of this reduction of tick infestations, the use of chemicals can be reduced, increasing their useful lifespan by delaying or eliminating the appearance of resistant tick strains and diminishing food contamination and environmental pollution.

It has become clear that a single method is not effective enough to achieve the control of tick infestations and ectoparasites in general. Overall tick control will depend on the harmonious integration of various methods instead of on one method alone, and vaccination can be included as the backbone of the adopted management strategy [36,77]. To date, the best results in tick control programs that include vaccination with Bm86 antigen have been obtained when governments have been involved and have applied regional implementation policies [36,68,77]. It has also been learned that these vaccines must be commercialized as a package which includes specialized technical support for training people in their proper use within these integrated programs. The main objective of these programs should be tick infestation control rather than eradication. With this tenet in mind, a clear strategy should be established from the beginning of a research project, when the proof-of-concept testing for new antigens against ticks will be performed in the laboratory. This strategy should guarantee a successful pipeline of technological development towards a marketable product which takes into consideration not only the efficacy needed for tick control in the field, but also novel methods for vaccine formulation that ensure the highest quality and longest lasting host immune response against the antigen, the industry’s demand for low cost, and consumer training in the correct implementation of the program, including vaccination. All these are imperative to enable effective development and commercialization of this innovative biotechnology and to make these products attractive to animal health companies.

Another significant reported impact on Bm86-vaccinated cattle is that the incidence of hemoparasitic diseases is decreased [36,37]. It is not clear whether this effect is caused by the reduction in tick infestations or because there is a specific effect of this antigen on the tick’s ability to transmit these pathogens, although there are preliminary studies that suggest the tick’s ability to act as a vector is affected by antibodies against Bm86 [78,79]. Another anti-tick antigen has also shown an ability to protect vaccinated hosts against viruses transmitted by infected ticks such as TBEV [80]. It appears at this point that an antigen design with a dual effect (against ticks and against tick-borne pathogens) could be one of the major impacts of a vaccine against ticks, taking into account that the main health concern about ectoparasites is their ability to transmit disease agents to host animals [81,82].

Finally, these successful results obtained in the practical implementation of vaccination against ticks could be extrapolated to the control of other ectoparasites such as the human body louse [83,84], mosquitoes [85,86,87], sand flies [88], and sea lice of salmonids [89,90], which can be affected in their life cycles via the same immunological mechanisms of a vaccinated host as have been described for ticks, and the control of which is currently addressed mainly through the use of chemicals.

## Figures and Tables

**Figure 1 pathogens-12-00231-f001:**
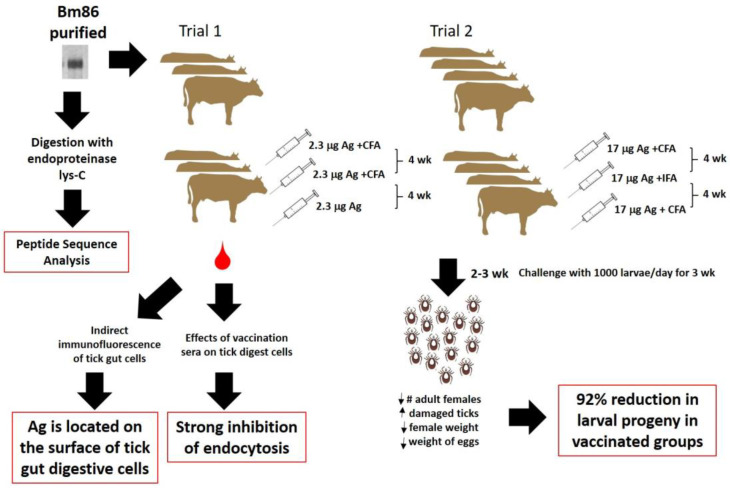
Scheme summarizing the experiments performed by Willadsen et al. in 1989 [25]. Legend: Ag—antigen; wk—weeks; CFA—complete Freund’s adjuvant; IFA—incomplete Freund’s adjuvant.

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
