# Peer review of "The Bm86 Discovery: A Revolution in the Development of Anti-Tick Vaccines"

_pathogens, 2023, doi:10.3390/pathogens12020231_

Round 1

Reviewer 1 Report

Rodriguez-Mellon describe the impacts of usage of an anti tick vaccine based on a midgut antigen. The experiments in this field from the past is referred, the possible outcome, future effects are discussed.

I suggest the publication of this manuscript as any data of methods for tick control worth discussing. Spreading these data among epidemiologists, parasitologists could be useful.

Although I still do not understand what kind of benefit we could achive from such vaccinations.

1., Vaccination can not decrease (prevent) the infection of hosts by various tick-borne pathogens. e.g. TBEV or pathogens of Lyme borreliosis.

2., In the environment 99% of ticks infest free-ranging species, mostly mammals, not our pets or animals of livestock species. We can not vaccinate free-ranging species, which maintain and fed, spread ticks.

A continuous survey on a certain territory, where ruminant livestock species are grazing should be carried out, and to check tick infestation of the vaccinated animals for years. Results of such an experiment would answer whether vaccination is really beneficial, or not.

In the paper non of these questions are metioned, although a correct picture should be shown about anti-tick vaccination for the readers.

Some remarks:

lines 93-94. – I do not understand what kind of side efffects the author means here. (ref 40.41) Please clarify.

-          direct designation of a vaccine is correct in a scientific paper of basic science?

-          lines 111-112. Similarly, what mean the author here by „control ticks”? Their number in the environment? Less number of questing ticks? More unsuccessful engorgements?

In temperate climate of Europe Ixodes ricinus is the overwhelming tick species in the environment. Is there any result about the impact of vaccination on this species? Should also be mentioned.

Author Response

I would like to thank you for your review to the manuscript. It is a valuable contribution to improve it. Taking in consideration your suggestions, I have sent a revised version of the manuscript in which the following issues have been addressed:
R1: …” I still do not understand what kind of benefit we could achieve from such vaccinations…”
In the introduction section, a first paragraph was included explaining the necessity to find new sustainable strategies to control ticks as alternative methods to the use of chemicals. In the “Impacts” section was also included an explanation about the most valuable effects of the vaccine inclusion in the tick control programs that can be expected at long-term.
R1: ... ” Vaccination cannot decrease (prevent) the infection of hosts by various tick-borne pathogens. e.g. TBEV or pathogens of Lyme borreliosis”.
In the “Impacts” section was included a commentary about the significant impact reported in Bm86 vaccinated cattle referred to the incidence diminution of hemoparasitic diseases and references to some studies suggesting a Bm86 role in the diminution of tick vector capacity were also included. In the same way, the demonstration of this inhibitory effect on TVEB transmission by other anti-tick antigen was also mentioned. These references give an idea about that the anti-tick vaccines could have also an effect on tick-borne pathogens. It must be evaluated with each specific antigen and each specific pathogen.
R1: … “In the environment 99% of ticks infest free-ranging species, mostly mammals, not our pets or animals of livestock species. We cannot vaccinate free-ranging species, which maintain and fed, spread ticks” and...” A continuous survey on a certain territory, where ruminant livestock species are grazing should be carried out, and to check tick infestation of the vaccinated animals for years. Results of such an experiment would answer whether vaccination is really beneficial, or not”.
These issues were not addressed in the manuscript because we consider that they are not specific for vaccination as a method to reduce tick infestations. They are valid not only for tick control by the immunological way but also for all strategies designed by humans to control ectoparasites. In general, all strategies including chemical, biological and immunological methods among others
are directed against ticks infesting livestock and companion animals. These tick infestations have a direct economic impact or have interest for public health. There are few examples in which some approaches have been designed to control ticks in wild life and these have been put into practice precisely due to direct effects on livestock. In the same way, it is known that there are cattle breeds more resistant to ticks than others and areas with climatic conditions in which there is a lesser tick infestation and it is not necessary to apply strategies in general to control ticks. In contrast, it is also known that many regions in tropical and subtropical areas would be unable to sustain economically viable livestock production without active tick control measures. Therefore, the
application of strategies in general to control ticks in a specific region is a result from a cost-benefit analysis.
R1: … “I do not understand what kind of side effects the author means here”
An unwanted effect of vaccines as skin irritation was mentioned in the text.
R1: … “direct designation of a vaccine is correct in a scientific paper of basic science?”
The name of the specific vaccine was eliminated from the text.
R1: …. “what mean the author here by “control ticks”? Their number in the environment? Less number of questing ticks? More unsuccessful engorgements?
One of the paragraph added to the “Impacts” section contains the explanation about the effects against ticks waited from host vaccination and their consequences for tick infestation control.
R1: …. “In temperate climate of Europe Ixodes ricinus is the overwhelming tick species in the environment. Is there any result about the impact of vaccination on this species? Should also be mentioned”.
As it is referred in the manuscript, there are many laboratory studies performed with different antigens against different tick species among which Ixodes ricinus is also included. References to some of these studies were incorporated in the manuscript as bibliographic references. However, as it is stated, all those studies have not gone beyond a proof of concept in experiments under controlled conditions with a few number of host animals. Data of anti-tick vaccine evaluation in field condition and its impact on a specific tick life cycle is only available for Bm86-based vaccines and against Rhipicephalus microplus ticks. For this reason, the impact of vaccination on Ixodes ricinus ticks is not mentioned.

Reviewer 2 Report

The commentary manuscript entitled ‘The Bm86 Discovery: A Revolution in the Development of Anti-Tick Vaccines’ by Rodriguez-Mallon is critically analysed and written.  The author summarizes the state-of-art of the development of anti-tick vaccines based on Bm86 antigen and emphasizes the impacts that the use of Bm86 based-formulations can lead to.

I accept this manuscript for its publication after minor revisions.

Below you find some minor comments to improve the manuscript.

-          Abstract: I would add a concluding sentence related to the impacts section

-          L 56: “5.6” and not “5 .6”

-          L 106: the term ‘don’t’ is very informal, please replace it with do not.

Author Response

I would like to thank you for the review of this manuscript. Your commentaries are a valuable contribution to improve it. Taking in consideration your suggestions, I have sent a revised version of the manuscript in which the following issues have been addressed:

R2: ... “Abstract: I would add a concluding sentence related to the impacts section”
It was added.
R2: ... “L 56: “5.6” and not “5 .6”
It was corrected.
R2: ... “L 106: the term ‘don’t’ is very informal, please replace it with do not”
It was corrected.